

# Evaluation of feature projection techniques in object grasp classification using electromyogram signals from different limb positions

Nantarika Thiamchoo and Pornchai Phukpattaranont

Department of Electrical Engineering, Faculty of Engineering, Prince of Songkla University, Hat Yai, Songkhla, Thailand

## ABSTRACT

A myoelectric prosthesis is manipulated using electromyogram (EMG) signals from the existing muscles for performing the activities of daily living. A feature vector that is formed by concatenating data from many EMG channels may result in a high dimensional space, which may cause prolonged computation time, redundancy, and irrelevant information. We evaluated feature projection techniques, namely principal component analysis (PCA), linear discriminant analysis (LDA), t-Distributed Stochastic Neighbor Embedding (t-SNE), and spectral regression extreme learning machine (SRELM), applied to object grasp classification. These represent feature projections that are combinations of either linear or nonlinear, and supervised or unsupervised types. All pairs of the four types of feature projection with seven types of classifiers were evaluated, with data from six EMG channels and an IMU sensors for nine upper limb positions in the transverse plane. The results showed that SRELM outperformed LDA with supervised feature projections, and t-SNE was superior to PCA with unsupervised feature projections. The classification errors from SRELM and t-SNE paired with the seven classifiers were from 1.50% to 2.65% and from 1.27% to 17.15%, respectively. A one-way ANOVA test revealed no statistically significant difference by classifier type when using the SRELM projection, which is a nonlinear supervised feature projection ($p = 0.334$). On the other hand, we have to carefully select an appropriate classifier for use with t-SNE, which is a nonlinear unsupervised feature projection. We achieved the lowest classification error 1.27% using t-SNE paired with a $k$-nearest neighbors classifier. For SRELM, the lowest 1.50% classification error was obtained when paired with a neural network classifier.

## INTRODUCTION

There were approximately 1.6 million persons with limb loss in the United States in 2005. It is projected that the total number of persons with limb loss will be 3.6 million by the year 2050 (*Ziegler-Graham et al., 2008*). A total of 35% of persons with limb loss (541 thousand persons) have an amputated upper extremity. Among the persons with upper limb loss, 41,000 were amputations above or below the elbow, and 500,000 were

Corresponding author
Pornchai Phukpattaranont,
pornchai.p@psu.ac.th

amputations of fingers or hands (*Ziegler-Graham et al., 2008*). In spite of the small fraction of upper limb amputations, these are disastrous incidents for the patients and cause disability preventing activities of daily living.

Prosthetic devices have been developed to enhance the quality of life for people with upper-limb amputation. Currently, the available commercial prostheses come under two general types, *i.e.* passive/cosmetic prosthesis and active/functional prosthesis. The cosmetic prosthesis is primarily designed to replace the missing parts of the body, though unable to function as a natural hand. In contrast, the functional prosthesis is mainly aimed at enabling the users to manipulate the device in performing activities of daily life (*Fang et al., 2015*; *Peerdeman et al., 2011*; *Cordella et al., 2016*; *Clement, Bugler & Oliver, 2011*; *Scheme & Englehart, 2011*; *Smirnov et al., 2021*).

A surface electromyogram (EMG) is a biological signal, which directly reflects nerves and muscle activities relating to body movements and can be recorded from the skin (*Scheme & Englehart, 2011*; *Liu, 2014*). Its noninvasive acquisition enables convenient uses in clinical applications (*Liu, 2014*). Therefore, the EMG signals from upper limb muscles are popularly used as a control source for the functional hand prostheses (*Anam & Al-Jumaily, 2020*; *Al-Angari et al., 2016*; *Radmand, Scheme & Englehart, 2016*; *Khushaba et al., 2016*; *Khairuddin et al., 2021*; *Feng et al., 2021*). Current control strategies of the functional hand prostheses are generally categorized into two groups, namely non-pattern recognition based and pattern recognition (PR) based control schemes (*Asghari Oskoei & Hu, 2007*). Prior publications reported the success of EMG-PR algorithm for hand prosthesis control with a high number of degrees-of-freedom (DOF) compared to other methods, such as on-off, proportional, direct, and finite state machine control (*Geethanjali, 2016*).

In the EMG-PR scheme, after the EMG signals from various muscles are acquired, features from each EMG channel are extracted. Then, a feature vector is formed by concatenating the features from all EMG channels. The successful classification accuracy for a high DOF hand prosthesis control requires either a large number of distinct features, such as autoregressive coefficients (*Anam & Al-Jumaily, 2020*; *Al-Angari et al., 2016*), spectral coefficients (*Khushaba et al., 2016*), and wavelet coefficients (*Al-Angari et al., 2016*; *Savareh et al., 2018*), or a large number of EMG channels (*Radmand, Scheme & Englehart, 2016*). However, a large number of EMG channels combined with a large number of features per channel may cause the feature vector to have an excessively high dimensionality. For example, in accordance with *Al-Timemy et al. (2013)*, 12 EMG channels were recorded in intact-limbed subjects to classify 15 hand and finger motions; and 11 EMG channels were acquired in amputee subjects to classify 12 hand and finger motions. Then, 11 features per EMG channel were extracted resulting in feature vectors with dimensions of 121 and 132.

A very high dimensional feature vector may cause some disadvantages such as long computation times. Also, redundant and irrelevant information in the high dimensional feature vector may degrade accuracy. For dimension reduction of the feature vector, feature projection is used (*Al-Timemy et al., 2013*; *Khushaba, Al-Ani & Al-Jumaily, 2010*; *Phukpattaranont et al., 2018*; *Chu et al., 2007*; *Chu, Moon & Mun, 2006*; *Liu, 2014*). Feature

projection attempts to construct a mathematical model, which transforms the significant information of a high-dimension space to a low-dimension space by analyzing the distribution of the reduced space. This allows for a visualizing of a high dimension space to a low dimension space (*Chu et al., 2007*). Feature projection can be categorized into two types, either linear or nonlinear. In addition, each type of feature projection can be further divided into either supervised or unsupervised. The most popular unsupervised linear feature projection is principal component analysis (PCA) (*Al-Timemy et al., 2013*; *Khushaba, Al-Ani & Al-Jumaily, 2010*; *Chu et al., 2007*; *Chu, Moon & Mun, 2006*; *Liu, 2014*; *Anam & Al-Jumaily, 2020*; *Rabin et al., 2020*). It is usually used as a baseline in comparison with other techniques.

For supervised linear feature projection, linear discriminant analysis (LDA) is the most popular (*Khushaba, Al-Ani & Al-Jumaily, 2010*; *Phukpattaranont et al., 2018*; *Chu et al., 2007*; *Lashgari & Maoz, 2021*). Its variants include an uncorrelated LDA, an orthogonal LDA, and a fuzzy LDA (*Khushaba, Al-Ani & Al-Jumaily, 2010*). The results from previous publications show that LDA has outperformed PCA (*Khushaba, Al-Ani & Al-Jumaily, 2010*; *Phukpattaranont et al., 2018*; *Chu et al., 2007*). For linearly separable data, the application of linear feature projection technique is appropriate. However, for non-linearly separable data, a nonlinear feature projection may give better performance. The supervised nonlinear technique used in spectral regression extreme learning machine (SRELM) was performed by *Phukpattaranont et al. (2018)* and *Anam & Al-Jumaily (2020)*. SRELM reduced the dimension of EMG feature vector from 66 to 13 in accordance with *Phukpattaranont et al. (2018)*. The accuracy from the classification of 14 hand and finger motions based on SRELM was superior to the linear projection techniques, *i.e.*, PCA, LDA, and LDA variants. According to *Anam & Al-Jumaily (2020)*, SRELM was evaluated for various number of hand motion classes (5–10 classes) and the various number feature dimensions (12–195 dimensions) using two EMG channels. Similar results were obtained. In other words, when the number of hand motion classes is 10, SRELM provided better classification results than LDA and spectral regression discriminant analysis (SRDA).

The t-distributed Stochastic Neighbor Embedding (t-SNE) is an unsupervised nonlinear feature projection technique. It has recently gained much interest and has been successfully used in many applications (*Devassy, George & Nussbaum, 2020*; *Exarchos et al., 2019*; *Oliveira, Machado & Andrade, 2018*). The t-SNE is widely used for visualizing hyperdimensional biosignals in two or three dimensions (*Birjandtalab, Pouyan & Nourani, 2016*; *Hajian, Etemad & Morin, 2019*; *Connan et al., 2016*). *Devassy, George & Nussbaum (2020)* applied the t-SNE to reduce the dimensionality of hyperspectral paper data. Its clustering performance was superior to PCA. For the dimension reduction of inertial and EMG data in the classification to distinguish healthy subjects from those suffering from Parkinson's disease, the t-SNE was compared to two other feature projections, PCA and Sammon's mapping, by *Oliveira, Machado & Andrade (2018)*. When the projected features were input to a support vector machine (SVM), classification accuracies for the test set from PCA, Sammon's mapping, and t-SNE were 67.8%, 74.1%, and 76.6%, respectively (*Oliveira, Machado & Andrade, 2018*).

Recently, several studies exposed the semantic gap between laboratory research and clinical implementation. Due to the assumption of EMG stationary, the traditional research examines the experiment in the laboratory environment, which is disregarded for dynamic and unintended movement. The EMG data was usually collected from a fixed position. In the realistic scenario, there are various factors that are able to impair the pattern of EMG signal, such as electrode displacement, muscle fatigue, as well as variation in force and limb position change (*Scheme & Englehart, 2011*; *Li, Shi & Yu, 2021*; *Simão et al., 2019*). These cause the transient change in the EMG signal, which degrades the performance of the classification and control system (*Scheme & Englehart, 2011*).

To overcome these limitations, the limb position change effects on EMG-PR were studied since the year of 2010 in *Scheme & Englehart (2011)*. The sensor fusion technique between EMG and various kinematic sensors, such as an accelerometer (ACC) (*Fougner et al., 2011*; *Shahzad et al., 2019*; *Geng, Zhou & Li, 2012*), gyroscope (GYR) (*Shahzad et al., 2019*), mechanomyogram (MMG) (*Geng, Zhou & Li, 2012*), and electromagnetic sensor (*Yang et al., 2017*; *Rabin et al., 2020*), were proposed to construct the limb position awareness system by training classifier with data from multiple limb positions. However, the integration of multiple sensors might bring about a high-dimensional feature vector.

In prior publications, feature projection techniques were evaluated with the EMG signals acquired from hand and finger motions when the limb positions were fixed (*Chu et al., 2007*; *Khushaba, Al-Ani & Al-Jumaily, 2010*; *Al-Timemy et al., 2013*; *Liu, 2014*; *Khushaba et al., 2014*; *Anam & Al-Jumaily, 2020*; *Rabin et al., 2020*). The classification of EMG signals from different limb positions is needed for the practical use of an EMG-based prosthetic hand in a real environment (*Shahzad et al., 2019*; *Jochumsen, Waris & Kamavuako, 2018*; *Yang et al., 2017*). This is more challenging compared to the classification of EMG signals from fixed limb positions because of the larger signal variations. Moreover, note that four types of feature projection techniques, namely unsupervised-linear, supervised-linear, unsupervised-nonlinear, and supervised-nonlinear, have not been compared with the same data. Therefore, in this paper, we propose to evaluate PCA, LDA, and SRELM in object grasp classification, using EMG and kinematics signals from different limb positions. We also include the evaluation of t-SNE, which is an unsupervised nonlinear feature projection technique that has provided successful results in other applications. However, it has not been reported in the context of recognition from EMG of hand and finger movements.

# DATA ACQUISITION AND EXPERIMENTAL PROTOCOL

## Sensor placement and data acquisition

This section describes sensor placement and data acquisition for EMG and the inertial measurement unit (IMU) signals. The details are as follows.

### EMG data acquisition

Before EMG data acquisition, the subjects cleaned their skin with alcohol for good signal quality. Then, six pairs of self-adhesive Ag/AgCl snap bipolar electrodes were placed on

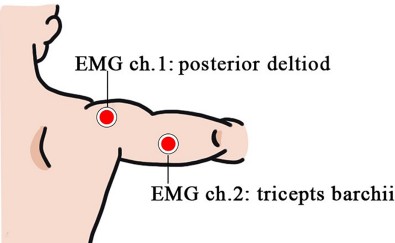

Right dorsal muscles

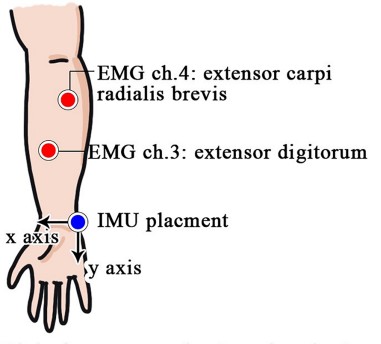

Right forearm muscles (anterior view)

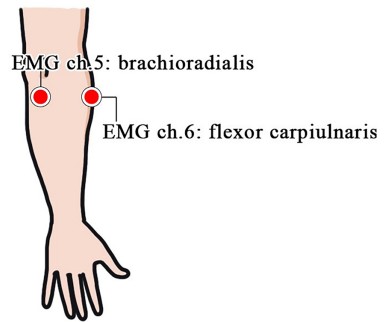

Right forearm muscles   (posterior view)

**Figure 1** **EMG and IMU sensor placement.**

**Table 1** **Summary of EMG and IMU sensor placements.**

| Sensor | Underlying muscle | Location |
| --- | --- | --- |
| EMG-CH1 | Posterior deltoid | Shoulder |
| EMG-CH2 | Triceps brachii | Upper arm |
| EMG-CH3 | Extensor pollicis longus | Forearm |
| EMG-CH4 | Extensor digitorum communis | Forearm |
| EMG-CH5 | Brachioradialis | Forearm |
| EMG-CH6 | Flexor digitorum profundus | Forearm |
| EMG-Ground | Not applicable | Left wrist |
| IMU | Not applicable | Right wrist |

three regions of the right arm, namely a pair on the posterior shoulder region, a pair on the upper arm, and four pairs on the forearm, as shown in Fig. 1. The six targeted muscles that correspond to this electrode placement are given in Table 1. All of the EMG signals were recorded by a commercial EMG measurement system (TMSI, MOBI) and were wirelessly transmitted to the computer using Bluetooth. The EMG signals were sampled at 1,024 Hz, amplified 1,000-fold, and filtered by a bandpass filter with a passband range from 20 to 500 Hz.

### IMU data acquisition

The kinematic signals were measured *via* an IMU sensor (MPU-6050TM) consisting of both a 3-axis accelerometer (ACC) and a 3-axis gyroscope (GYR) integrated on a single chip. The IMU sensor was attached on the right wrist with the direction of the *Y*-axis

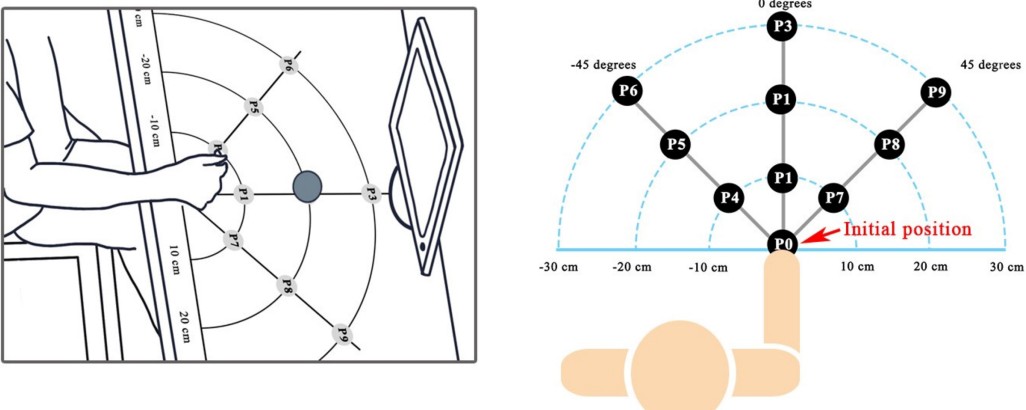

**Figure 2 Experimental setup and nine positions for object placement.**

pointing out forward and the direction of *X*-axis pointing into the center of the subject's body. A microcontroller (Arduino Uno) was used to record the IMU signal with a sampling rate of 60 Hz. A trigger signal was generated so that the EMG and IMU data were recorded simultaneously.

## Experimental protocol

Six-channel EMG signals and the signals from an IMU during object grasps were collected from 14 healthy subjects (seven males and seven females) aged between 18 and 44, who were not reported of neuromuscular disorder history. All subjects understood the experimental protocol and gave the written informed consent before performing experiment. The experiments were in accordance with the Declaration of Helsinki and duly approved by the Human Research Ethical Committee of the Faculty of Medicine, Prince of Songkla University (REC.62-310-19-2).

After attaching the EMG and IMU sensors, the participants performed the experiment by sitting in front of a grid table with 90 degrees of elbow flexed. Their right wrist was over position P0 as shown in Fig. 2. Five object grasps were studied consisting of a sphere (diameter 8 cm), a cylinder (diameter 3.3 cm), a keycard (thickness 0.15 cm), an eraser (thickness 1.4 cm), and a pen (diameter 1 cm). We examined the effects of limb position change by placing the objects at nine positions (P1–P9), which were divided into main three orientations, *i.e.*, (1) the middle direction (P1–P3), (2) 45 degrees apart from the middle direction to the left (P4–P6), and (3) 45 degrees to the right (P7–P9), as shown in Fig. 2.

A demonstration video was shown on a monitor during data collection for guiding the participant's movements and to control the timing. There were three steps in grasping an object placed on one of the nine positions, as follows: Step (1) reach to grasp the target object within 1–3 s depending on how far the target position is from the initial position P0, Step (2) grasp and lift the target object at the target position for 2 s, and Step (3) place the object to the target position and return to P0 within 1–3 s. All participants were asked to perform five repetitions for each grasp type and each object position. As a result, in total

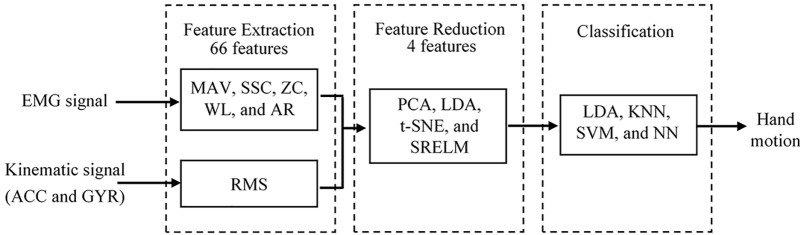

**Figure 3 Schematic of proposed analytical method.**

225 trials (5 objects × 9 positions × 5 repetitions) were performed. To avoid fatigue, the participants took a rest for 2–5 min after finishing the experiment in each trial.

# PROPOSED ANALYTICAL METHOD

Figure 3 shows the schematic of proposed analytical method consisting of feature extraction, feature projection, and classification. Details are as follows.

## Feature extraction

Six channels of EMG signals from the shoulder and upper limb muscles were processed with two digital filters. Firstly, to suppress motion artifacts, a fifth-order Butterworth bandpass filter with a passband of 20–450 Hz was employed. Secondly, we eliminated the power line interference at 50 Hz by a notch filter. In addition, the IMU signals were smoothed with a moving average filter for reduction of the signal fluctuations. Then, the filtered EMG and IMU signals from the step "grasp-and-lift object at the target position for 2 s" were segmented into short sessions for further analysis. The size of each session was 150 ms with a 50 ms increment for obtaining the next segmented session with 100 ms overlap on the current one.

To identify the grasp type, the well-known time-domain features extracted from the EMG signal $u_i$ in each session (150 ms) were mean absolute value (MAV), zero crossings (ZC), waveform length (WL), slope sign changes (SSC), and the sixth order autoregressive coefficients (AR). The brief details on each feature calculation are as follows.

- MAV is determined by taking the absolute value of all EMG signals in each session and averaging them, which can be expressed as

$$\text{MAV} = \frac{1}{N} \sum_{i=1}^{N} |u_i|, \tag{1}$$

where $u_i$ is the EMG amplitude at index $i$ and $N$ is the size of each EMG session.

- ZC denotes the number of times the EMG amplitude changes sign in each session, which can be defined as

$$\text{ZC} = \sum_{i=1}^{N-1} [g(u_i \times u_{i+1})],$$
$$\text{where } g(d) = \begin{cases} 1, & \text{if } d < 0 \\ 0, & \text{otherwise} \end{cases} \tag{2}$$

- WL is calculated by averaging the absolute value of the differences between each subsequent EMG pair in each session, which is given by

$$\text{WL} = \sum_{i=1}^{N-1} |u_{i+1} - u_i|. \tag{3}$$

- SSC is the number of times the slope of the EMG amplitude changes sign. It is defined as

$$\text{SSC} = \sum_{i=2}^{N-1} [h\{(u_i - u_{i+1})(u_i - u_{i-1})\}],$$
$$\text{where } h(d) = \begin{cases} 1, & \text{if } d > 0 \\ 0, & \text{otherwise} \end{cases} \tag{4}$$

- AR represents each EMG signal sample as a linear combination of the preceding sample and a white noise error component $\varepsilon_i$, which can be given by

$$u_i = \sum_{p=0}^{P-1} a_p u_{i-p} + \varepsilon_i, \tag{5}$$

where $a_1, a_2, \ldots, a_P$ are the feature values from the model and $P$ is the order of the AR model, which is 6 in this paper.

All feature values from each EMG session are concatenated to form a feature vector with a dimension of 60 (10 features per channel × 6 EMG channels). Furthermore, six root-mean square (RMS) values were determined for each IMU session (3 from ACC and 3 from GYR). The RMS can be expressed as

$$\text{RMS} = \sqrt{\frac{1}{M} \sum_{i=1}^{M} v_i^2}, \tag{6}$$

where $v_i$ is the IMU amplitude at index $i$ and $M$ is the size of each IMU session. A feature vector with a dimension of 6 is obtained by combining all IMU feature values.

We concatenated the EMG feature vector with the IMU feature vector to achieve the combined feature vector with a length of 66 to represent an analytical session window. All feature vectors from each participant were randomized and separated into training and testing datasets using five-fold cross validation. In other words, for each subject, data from all recorded positions were divided into five sections. When one of the five sections was evaluated as the test data (20%), the remaining sections were the training data (80%). The classification error was obtained by averaging the errors from all five test sections. All processing was performed on a notebook (AMD Ryzen 7 3750H CPU and 8 GB of RAM) using *MATLAB (2021)*.

Before feature projection, the minimum and maximum values of MAV, ZC, WL, SSC, AR from 6 EMG channels and the minimum and maximum values of RMS from 3-asix ACC and GYR from the training data were determined and kept. These values were used to normalize both the training and testing datasets into the range from −1 to 1.

## Feature projection

A high-dimensional feature vector might cause high computational cost. A feature projection technique was used to map the original feature vector into a small feature space, which could save in required computing resources. In this paper, the four feature projection techniques PCA, LDA, SRELM, and t-SNE were evaluated and compared. It should be noted that the dimensionality of each projected feature set after feature projection, was four (the number of grasp type minus one).

A set of $n$ feature vectors $\mathbf{x}_i \in R^m$ in the high dimensional space can be written as an $n \times m$ feature matrix $\mathbf{X} = [\mathbf{x}_1, \mathbf{x}_2, \ldots, \mathbf{x}_n]^T$, where $n$ is the total number of feature vectors. After feature projection, we obtain the low dimensional counterpart $\mathbf{y}_i \in R^k$ ($k < m$) of the high dimensional feature vector $\mathbf{x}_i$. Then, the corresponding low dimensional $n \times k$ feature matrix can be expressed as $\mathbf{Y} = [\mathbf{y}_1, \mathbf{y}_2, \ldots, \mathbf{y}_n]^T$. The brief details on each projection technique are as follows.

- PCA is based on the concept of finding the orthogonal basis vectors, which maximize variance. After the eigenvalues and eigenvectors are determined from the eigenvalue decomposition of a covariance matrix $\mathbf{X}^T\mathbf{X}$, an $m \times k$ projection matrix $\mathbf{U}_k$ is constructed using the eigenvectors with the $k$ highest eigenvalues (*Martinez & Kak, 2001*). Then, the feature vectors in low dimensional space are obtained by projecting $\mathbf{X}$ onto $\mathbf{U}_k$ ($\mathbf{Y} = \mathbf{X}\mathbf{U}_k$).

- LDA aims to transform the data by maximizing the ratio of the between-class variance to the within-class variance. The within-class variance is given by *Tharwat et al. (2017)*

$$\mathbf{S}_w = \sum_{j=1}^{c} \sum_{i=1}^{n_j} (\mathbf{x}_i^j - \mu_j)(\mathbf{x}_i^j - \mu_j)^T, \tag{7}$$

where $\mathbf{x}_i^j$ is the $i$th feature vector of class $j$, $\mu_j$ is the mean vector of class $j$, $c$ is the number of classes, and $n_j$ the number of feature vectors in class $j$. The between-class variance can be expressed as *Tharwat et al. (2017)*

$$\mathbf{S}_b = \sum_{j=1}^{c} n_j(\mu_j - \mu)(\mu_j - \mu)^T, \tag{8}$$

where $\mu$ represents the mean of all classes. An $m \times k$ projection matrix $\mathbf{V}_k$, which maximizes the ratio of between-class and within-class variability, can be obtained from the eigenvectors of $\mathbf{S}_w^{-1}\mathbf{S}_b$. In other words, after $\mathbf{S}_w^{-1}\mathbf{S}_b$ is decomposed using eigenvalue decomposition, $\mathbf{V}_k$ is constructed using the eigenvectors with the $k$ highest eigenvalues. Then, the feature vectors in low dimensional space are obtained by projecting $\mathbf{X}$ onto $\mathbf{V}_k$ ($\mathbf{Y} = \mathbf{X}\mathbf{V}_k$).

- SRELM was evaluated for its projection performance with features extracted using the EMG signals acquired from a fixed limb position (*Phukpattaranont et al., 2018*). It is composed of two main algorithms, *i.e.*, extreme learning machine (ELM) and spectral

| Table 2 Pseudo-code describing the SRELM projection algorithm. |
|---|
| **Algorithm 1: SRELM feature projection in the training stage** |
| **Input:** the number of hidden nodes $L$, the alpha parameter $\alpha$, the training feature vectors $\mathbf{X}_{tr} = [\mathbf{x}_1, \mathbf{x}_2,...,\mathbf{x}_n]^T \in R^{n \times m}$, and the training class label vector $\mathbf{t} \in R^n$ |
| **Output:** the output weight $\mathbf{U} = [\mathbf{u}_1, \mathbf{u}_2,...,\mathbf{u}_{(c-1)}] \in R^{L \times (c-1)}$ |
| 1) Generate randomly hidden node parameters $(\mathbf{a}_i, b_i), i = 1, 2, \ldots, L$ |
| 2) Calculate the hidden layer output matrix $\mathbf{H} = [\mathbf{h}(\mathbf{x}_1), \mathbf{h}(\mathbf{x}_2),...,\mathbf{h}(\mathbf{x}_n)]^T \in R^{n \times L}$, where $\mathbf{h}(\mathbf{x}) = [g(\mathbf{a}_1 \cdot \mathbf{x}_1 + b_1), \mathbf{a}_2 \cdot \mathbf{x}_2 + b_2) \ldots g(\mathbf{a}_L \cdot \mathbf{x}_L + b_L)]^T$ and $$g(\mathbf{a}_i \cdot \mathbf{x} + b_i) = \frac{1}{1 + e^{\mathbf{a}_i \cdot \mathbf{x} + b_i}}$$ |
| 3) Calculate an orthogonal matrix $\mathbf{Z} = [\mathbf{z}_1, \mathbf{z}_2,...,\mathbf{z}_{(c-1)}] \in R^{n \times (c-1)}$ from QR decomposition of the class label vector $\mathbf{t}$ |
| 4) Calculate the output weight $\mathbf{U}$ which subjects to $\mathbf{Hu} = \mathbf{z}$ from the solution of regularized least squares problem: $$\mathbf{u} = \arg \min_{\mathbf{u}} \left( \sum_{i=1}^{n} \left( \mathbf{u}^T \mathbf{h}(\mathbf{x}_i) - z_i \right)^2 + \alpha \sum_{j=1}^{L} u_j^2 \right)$$ |
| **Algorithm 2: SRELM feature projection in the testing stage** |
| **Input:** the testing feature vectors $\mathbf{X}_{ts} = [\mathbf{x}_1, \mathbf{x}_2,...,\mathbf{x}_{ns}]^T \in R^{ns \times m}$ and the output weight $\mathbf{U} = [\mathbf{u}_1, \mathbf{u}_2,...,\mathbf{u}_{(c-1)}] \in R^{L \times (c-1)}$ from the training stage |
| **Output:** the projected feature vectors $\mathbf{Y} = [\mathbf{y}_1, \mathbf{y}_2,...,\mathbf{y}_{ns}]^T \in R^{ns \times (c-1)}$ |
| 1) Generate randomly hidden node parameters $(\mathbf{a}_i, b_i), i = 1, 2, \ldots, L$ |
| 2) Calculate $\mathbf{H}_{ts} = [\mathbf{h}(\mathbf{x}_1), \mathbf{h}(\mathbf{x}_2),...,\mathbf{h}(\mathbf{x}_{ns})]^T \in R^{ns \times L}$ |
| 3) Calculate the projected features from $\mathbf{Y} = \mathbf{H}_{ts}\mathbf{U}$ |

regression (SR). While the ELM is employed unsupervised, the labels are included in the SR. As a result, SRELM is exploited as a supervised dimensionality reduction. Pseudocode describing the SRELM projection algorithm is shown in Table 2.

There are two parameters in SRELM calculation, *i.e.*, the number of hidden nodes and alpha. A grid search strategy was used to determine both. The alpha was varied from 1 to 10 with an increase of 1, while the number of hidden nodes was varied from 50 to 1,500 with an increment of 50 nodes. The best parameters were chosen from the combinations that produced the lowest classification error. As a consequence, alpha 1 was employed and 1,000 hidden nodes were used.

- t-SNE is an unsupervised nonlinear dimensionality reduction technique, which is introduced by *Maaten & Hinton (2008)*. In t-SNE, the similarity of feature vectors $\mathbf{x}_i$ and $\mathbf{x}_j$ in high dimensional space is determined using the conditional probability $p_{j|i}$, which is defined as *Maaten & Hinton (2008)*

$$p_{j|i} = \frac{e^{-\|\mathbf{x}_i - \mathbf{x}_j\|^2 / 2\sigma_i^2}}{\sum_{k \neq i} e^{-\|\mathbf{x}_i - \mathbf{x}_k\|^2 / 2\sigma_i^2}}. \tag{9}$$

The feature vectors from the same grasp type give relatively high $p_{j|i}$ because of the similarity between $\mathbf{x}_i$ and $\mathbf{x}_j$. Then, to obtain a symmetric matrix, the joint probability $p_{ij}$ is determined by *Maaten & Hinton (2008)*

$$p_{ij} = \frac{p_{i|j} + p_{j|i}}{2n}. \tag{10}$$

For the low dimensional counterparts of the high dimensional feature vectors, a joint probability $q_{ij}$ can be expressed as *Maaten & Hinton (2008)*

$$q_{ij} = \frac{\left(1 + \|\mathbf{y}_i - \mathbf{y}_j\|^2\right)^{-1}}{\sum\limits_{k \neq l}\left(1 + \|\mathbf{y}_k - \mathbf{y}_l\|^2\right)^{-1}}. \tag{11}$$

t-SNE aims to find a low-dimensional data representation that minimizes the mismatch between $p_{ij}$ and $q_{ij}$. This can be achieved by minimizing a cost function $C$ determined based on a single Kullback–Leibler divergence using a gradient descent method, which is given by *Maaten & Hinton (2008)*

$$C = \sum_i \sum_j \left(p_{ij} \log \frac{p_{ij}}{q_{ij}}\right). \tag{12}$$

## Classification

After feature projection, we evaluate the classification error of projected features from each feature projection technique. Seven linear and nonlinear classifiers, which have been employed for EMG pattern classification of hand and finger movements in prior publications, are tested and compared in this paper. The linear classifiers, which decide a class based on a linear combination of feature values, consist of LDA, naive Bayes (NB), and SVM with a linear kernel (SVML). On the other hand, four nonlinear classifiers include k-nearest neighbors (KNN), SVM with a radial basis function kernel (SVMB), SVM with a polynomial kernel (SVMP), and neural network (NN).

While parameter selection for LDA and NB was not needed, the parameters for KNN, SVM, and NN were determined by a grid search strategy. The parameter k in KNN was set to 3, resulting from the best selection in the range from 1 to 7. For SVM, gamma and cost parameters are varied in a range between 0.5 and 6 with a 0.5 interval. The optimal parameters were picked from those that resulted in the lowest classification error. As a result, gamma in the kernel function was set to one and cost was also set to one. For NN, three-layered feed-forward back-propagation neural networks consisting of four neurons in the input layer, 14 neurons in the tan-sigmoid hidden layer, and five neurons in the linear output layer were used. The number of neurons in the input and output layers was fixed by the dimension of the input feature vector and the number of grasp types, respectively. On the other hand, the number of neurons in the hidden layer 14 was determined from a grid search with a range between 1 and 50.

### Projected feature evaluation

To quantify the pattern characteristics of the projected features, we employ separability index (SI), mean semi-principal axis (MSA), and repeatability index (RI). The SI is defined as a half of the modified Mahalanobis distance, and is given by *Nilsson, Håkansson & Ortiz-Catalan (2017)*.

$$\text{SI} = \frac{1}{c}\sum_{i=1}^{c} \min_{j=1,\dots i-1,i+1,\dots c} \frac{1}{2}\sqrt{\left(\mu_i - \mu_j\right)^T \mathbf{S}^{-1}\left(\mu_i - \mu_j\right)}. \tag{13}$$

where $\mathbf{S} = \dfrac{\mathbf{S}_i + \mathbf{S}_j}{2}$ is the average covariance matrix, $\mathbf{S}_i$ is the covariance matrix of class $i$, and $\mathbf{S}_j$ is the covariance matrix of class $j$. The SI value indicates quantitative measurement of grasp cluster separation. The higher SI indicates the better grasp cluster separation.

The MSA measures the compactness of feature distribution in each cluster. Better cluster compactness leads to a lower MSA. It is determined from the geometric mean of the semi-principal axes averaged across the five grasp types, which is defined as

$$\text{MSA} = \frac{1}{c}\left(\sum_{i=1}^{c}\left(\prod_{j=1}^{k} a_{ij}\right)^{\frac{1}{k}}\right), \tag{14}$$

where $a_{ij}$ is from the $k$ highest eigenvalues.

The RI measures the ability to reproduce the features determined from one position to the others. It is determined as one-half the Mahalanobis distance between the mean feature vector for a training dataset and a testing dataset, averaged across the nine positions and the five grasp types as given by

$$\text{RI} = \frac{1}{c}\sum_{i=1}^{c}\frac{1}{2}\sqrt{\left(\mu_{Tri} - \mu_{Tsi}\right)^T \mathbf{S}_{Tri}^{-1}(\mu_{Tri} - \mu_{Tsi})}, \tag{15}$$

where $\mathbf{S}_{Tri}$ is the covariance of the $i$th-class training data, $\mu_{Tri}$ and $\mu_{Tsi}$ are the mean vectors of the $i$th-class training and testing data, respectively. The better consistency in feature characteristics from different positions leads to a lower RI.

## RESULTS

### Projected feature evaluation

In this section, we use SI and MSA values to assess the projected features from PCA, LDA, t-SNE, and SRELM in terms of grasp cluster separation and grasp cluster compactness. Table 3 shows average SI and MSA values from the four feature projection techniques. The mean of SI from SRELM (14.88) is the highest compared to those from PCA (6.42), LDA (11.05), and t-SNE (9.02). In other words, the projected features from SRELM give the best cluster separation for five different grasps compared with other feature projection techniques. In terms of cluster compactness, LDA is the best, which is supported by an MSA of 0.0015.

To gain more insight on the resulting SI and MSA, we compare scatter plots of the MAV and SCC feature vectors before feature projection from all placement positions in Fig. 4

**Table 3 Average values of SI and MSA from four feature projection techniques.**

| Projection type | SI | MSA |
|---|---|---|
| PCA | 6.42 | 0.0164 |
| LDA | 11.05 | 0.0015 |
| t-SNE | 9.02 | 0.0327 |
| SRELM | 14.88 | 0.0082 |

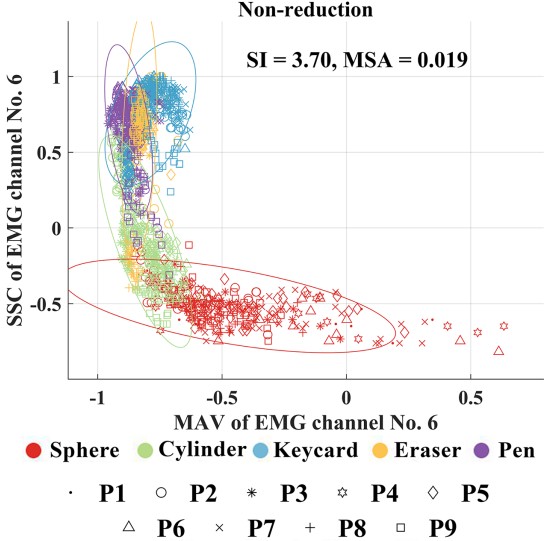

**Figure 4 Scatter plots of the MAV and SCC feature vectors before feature projection from all placement positions.** While the features from a sphere, a cylinder, a keycard, an eraser, and a pen are shown using red, green, blue, orange, and magenta colors, respectively, the features from positions 1 to 9 are shown using point, circle, asterisk, hexagram, diamond, triangle, cross, plus sign, and square markers, respectively.                     

with those of the first two elements of the projected feature vectors when using the four feature projection techniques (PCA, LDA, t-SNE, and SRELM) in Fig. 5. We can see that the degree of cluster separation in the scatter plot from the SRELM is noticeable compared to the others. Moreover, the SI of SRELM at 11.39 is also in line with its separation degree, which is higher than that from LDA (SI = 6.99). On the other hand, PCA and t-SNE provide the small degree of cluster separation resulting in the SI at 4.60, and 2.37, respectively.

In addition, the cluster compactness of SRELM (MSA = 0.0014) is the best, superior to PCA (MSA = 0.0030), LDA (MSA = 0.0017), and t-SNE (MSA = 0.0066). The visualization of cluster sizes from four feature projection techniques shown in Fig. 5 coincides with the reported MSA values.

## Classification accuracy

Classification error rates from all pairwise combinations of the four feature projection techniques with the seven classifiers are shown in Fig. 6. The classification errors from

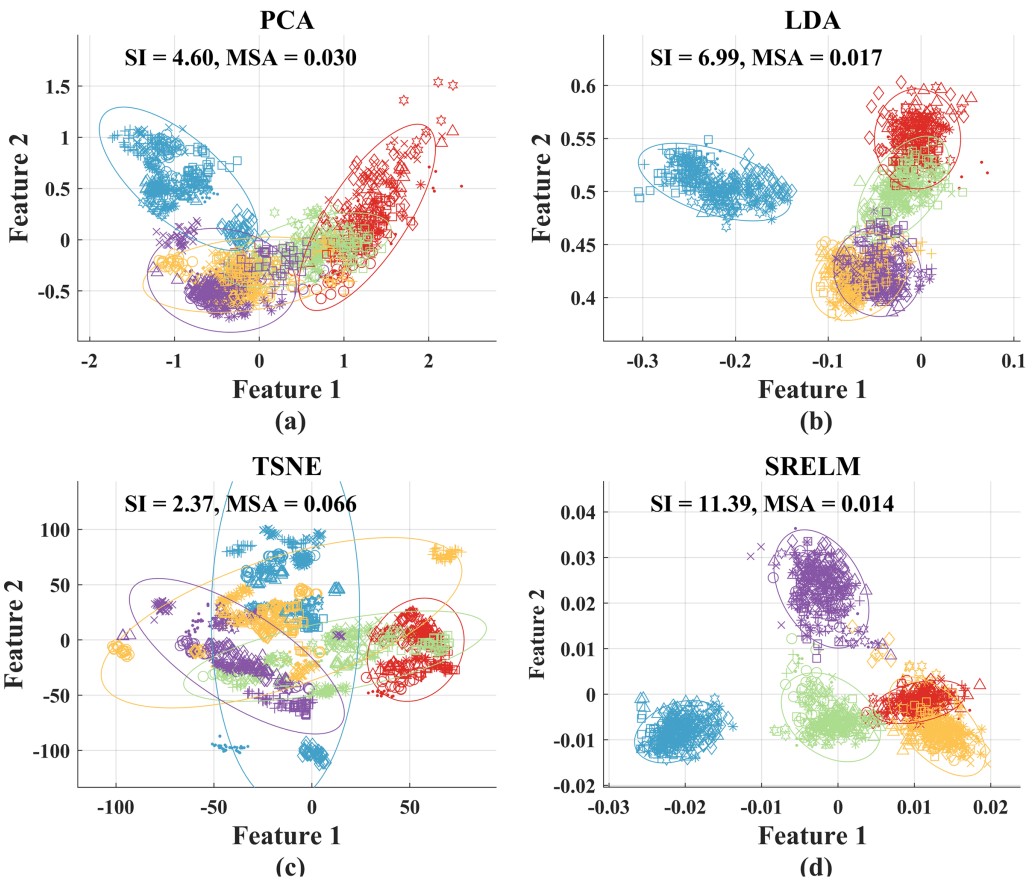

**Figure 5 Scatter plots of the first two elements of the reduced feature vectors from all placement positions when using four feature projection techniques: (A) PCA, (B) LDA, (C) t-SNE, and (D) SRELM.** While the features from a sphere, a cylinder, a keycard, an eraser, and a pen are shown using red, green, blue, orange, and magenta colors, respectively, the features from positions 1 to 9 are shown using point, circle, asterisk, hexagram, diamond, triangle, cross, plus sign, and square markers, respectively.                              

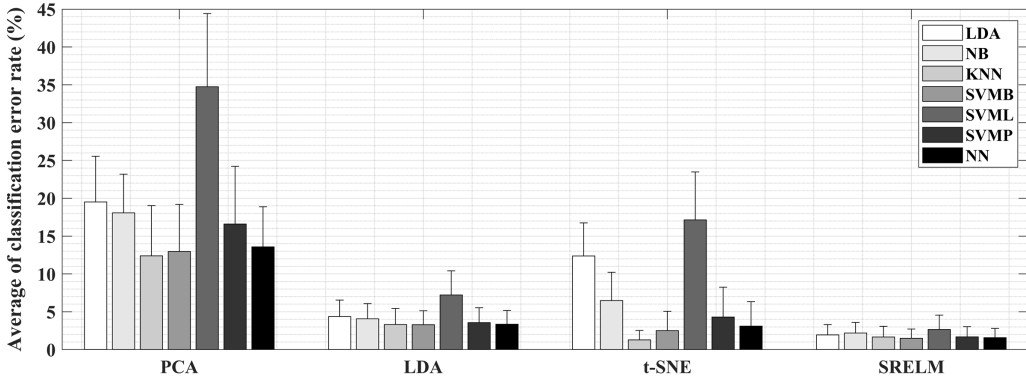

**Figure 6 Mean and standard deviation of classification errors from all pairwise combinations of four feature projection techniques and seven classifiers.**   

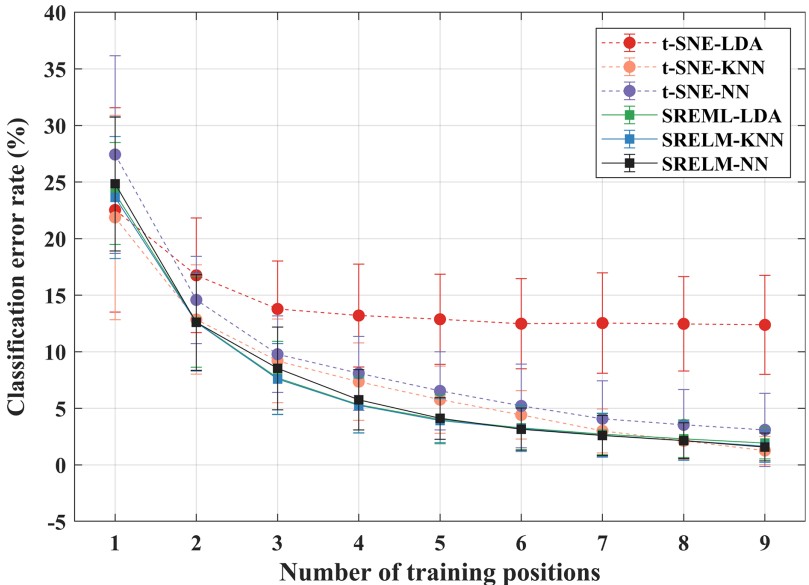

**Figure 7 The average classification error rate for difference schematics of feature projection and classifier when reducing availability of training positions.** The result is averaged over all 14 subjects, five classes, and nine testing positions.

SRELM are lower than from the others. In other words, the mean classification errors when using SRELM with the classifiers LDA, NB, KNN, SVMR, SVML, SVMP, and NN are 1.93%, 2.18%, 1.65%, 1.50%, 2.65%, 1.67%, and 1.58%, respectively. They are lower than those from LDA projection, which are in the range from 3.28% to 7.22%. The mean classification errors from t-SNE are minimum at 1.27% when paired with KNN classifier and maximum at 17.15% when paired with SVML classifier. Moreover, the mean classification errors from PCA are in the range from 12.39% to 34.74%, which are inferior to the others.

Two-way ANOVA analysis with a significance level of 0.05 was employed to investigate the effects of choices of projection technique and of classifier on the classification error rate. The results reveal that there is a significant difference when using these four projection techniques ($p < 0.001$). In addition, the *post hoc* test using Fisher's Least Significant Difference (LSD) shows that the classification error rates from LDA and SVML classifiers are significantly higher than those of the other classifiers ($p < 0.019$), while there are no significant differences among NB, KNN, SVMB, SVMP and NN classifiers. However, a one-way ANOVA test reveals that there is no statistically significant difference by classifier type when using the SRELM projection ($p = 0.334$).

Figure 7 shows comparison of average classification error rate from the six pairs of two feature projection techniques (t-SNE and SRELM) with three classifiers (LDA, KNN, and NN) as functions of training position count. The classification error rate declines when the number of training positions increases, in all cases of those six pairs. The results from t-SNE-LDA provide a noticeably higher error compared to the other pairs. A three-way

**Table 4 The average and standard deviation of classification error rate across 14 subjects for different numbers of training positions by SRELM feature projection and NN classifier.**

| Training position(s) | CER ± SD (%) | SI | RI |
|---|---|---|---|
| P5 | 24.84 ± 5.93 | 8.29 | 3.52 |
| P5–P9 | 12.60 ± 4.23 | 9.88 | 1.19 |
| P1–P5–P9 | 8.53 ± 3.66 | 10.97 | 0.67 |
| P1–P3–P5–P9 | 5.75 ± 2.66 | 12.09 | 0.44 |
| P1–P3–P5–P8–P9 | 4.11 ± 1.84 | 12.98 | 0.29 |
| P1–P3–P4–P6–P7–P9 | 3.16 ± 1.86 | 13.59 | 0.20 |
| P1–P2–P3–P4–P6–P7–P9 | 2.60 ± 1.78 | 14.00 | 0.15 |
| P1–P2–P3–P4–P5–P6–P7–P9 | 2.15 ± 1.58 | 14.58 | 0.11 |
| P1–P2–P3–P4–P5–P6–P7–P8–P9 | 1.58 ± 1.23 | 15.16 | 0.06 |

ANOVA revealed that there is a significant difference for interaction between all three factors: projection type, classifier type, and the number of training positions ($p = 0.001$). Moreover, all three factors significantly affect the classification error rate. The *post hoc* analysis with LSD indicates that the projection techniques and classifiers are significantly different for every pair, whereas there is no significant difference when the number of training positions is greater than or equal to 7.

To investigate the effects of classifier type with different projection techniques, the mean values of classification errors from applying t-SNE and SRELM were compared by a two-way ANOVA analysis over classifier type and the number of training positions as the two factors. The significant results were subjected to LSD *post hoc* test with 0.05 significance level.

Considering the result from t-SNE pairs, the rate of declination depends on the classifier type ($p < 0.001$). For example, there is no statistically significant difference in classification error rate when the number of training positions is greater than three in t-SNE-LDA pairs ($p = 0.467$), which is different from t-SNE-KNN (the number of training positions is greater than seven, $p = 0.264$) and t-SNE-NN (the number of training positions is greater than six, $p = 0.212$) pairs. On the other hand, SRELM provides a similar declination trend for all three pairs ($p = 0.704$). In other words, there is no statistically significant difference in classification error rate when the training positions is greater than six in SRELM ($p > 0.191$).

Table 4 presents the best positions used in training step corresponding to each number of training positions from SRELM-NN pair shown in Fig. 7. For example, the best six training potions are P1, P3, P4, P6, P7, and P9, which give an average classification error rate of 3.16%. When the number of training positions increases from 7 to 9, the corresponding average classification error rates are 2.60%, 2.15%, and 1.58%, respectively, which are not significantly different. When the number of training positions increases from one to nine, the RI decreases from 3.52 to 0.06. These results indicate that degree of overlap between training and testing data increases when the number of training positions increases as shown by decrease in RI.

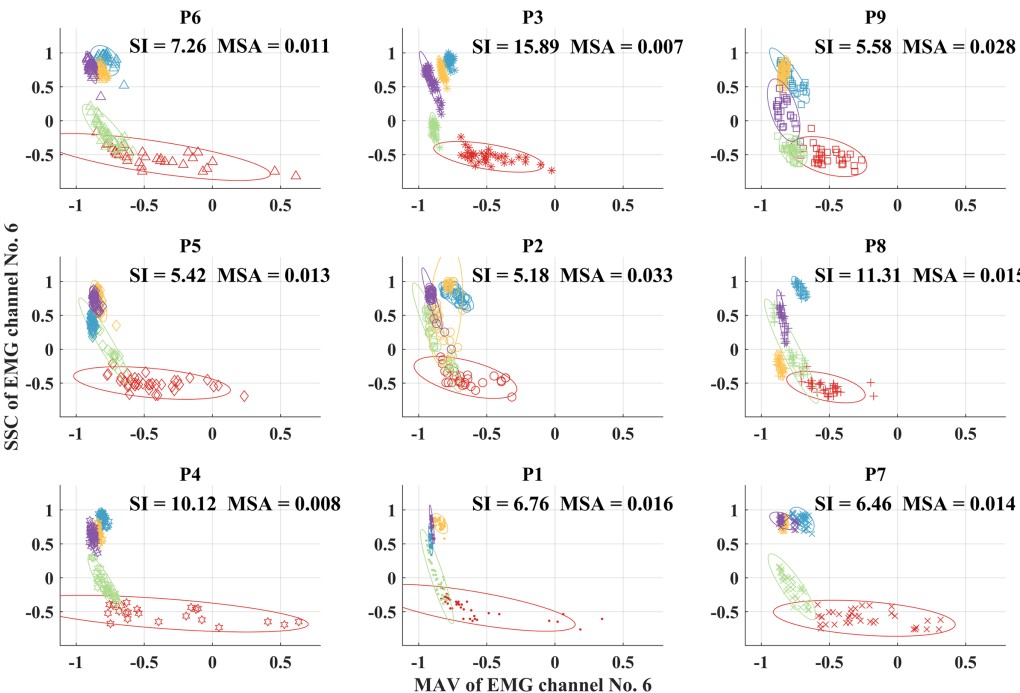

**Figure 8 Scatter plots of the MAV and SCC feature vectors for each placement position.**

## DISCUSSION

### Projected feature evaluation

This study examined the effects of projection technique and classifier type on hand grasp classification during variation in limb positions. Our results indicate that the variation in limb positions causes feature distribution changes, resulting in classification performance impairment.

Figure 8 shows the scatter plots between the MAV and the SCC feature vectors for each placement position before processing with feature projection. The clusters of feature vectors representing each grasp type in each position are quite overlap. Moreover, it is seen that the centers of clusters from each grasp type change when the positions change, resulting in SI values from 5.18 to 15.89. When all clusters from all positions are combined, the degree of cluster overlap increases resulting in the degradation of SI to 3.70 as shown in Fig. 4.

This was also reported in a prior publication. *Jochumsen, Waris & Kamavuako (2018)* investigated the distributions of EMG features from different motion classes with limb position changes. The results showed that shifts in mean values and standard deviations of some motion classes occurred when changing limb positions (*Jochumsen, Waris & Kamavuako, 2018*). *Fougner et al. (2011)* explained that muscles need to adjust the contractile strength resulting in changes of muscle shape, length and the number of active motor units, to stabilize the limb during performing specific hand motions at the different limb positions. This variation brought about transitions in amplitude and frequency of EMG characteristics.

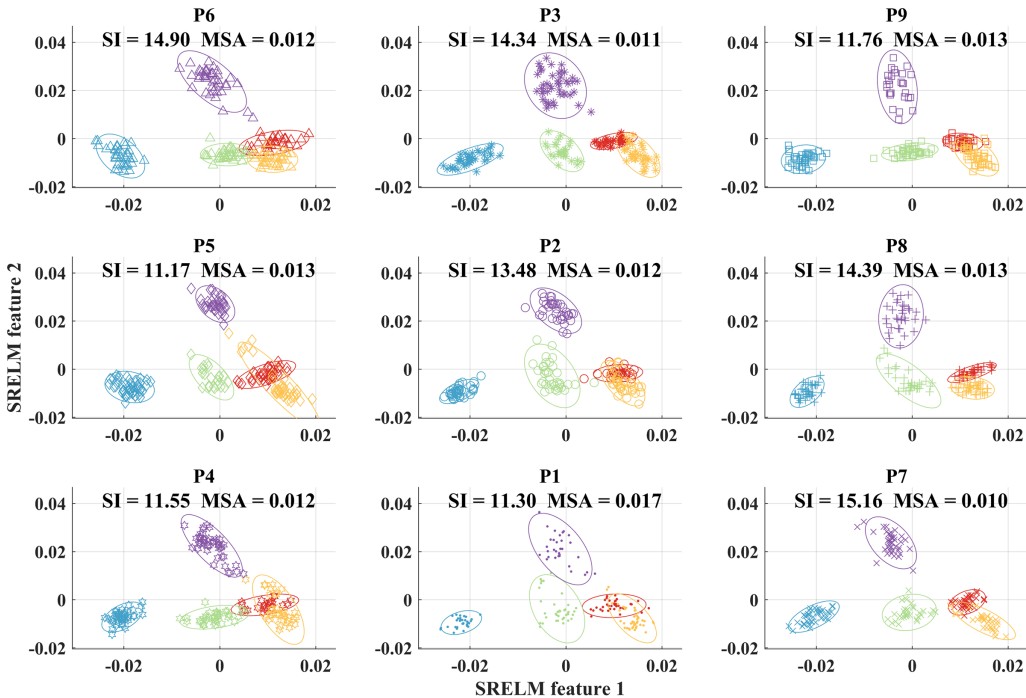

**Figure 9** Scatter plots of the first two elements of the reduced feature vectors from each placement position after applying SRELM feature projection.

In addition, it was reported that the MAV feature was more resistant to limb position changes compared to WL, ZC, and SSC features (*Jochumsen, Waris & Kamavuako, 2018*). The pairing of MAV and WL was suggested as the best feature source for surface EMG signals (*Jochumsen, Waris & Kamavuako, 2018*). However, our results showed that the best feature pair was MAV and SSC.

For comparison, we show the scatter plots between the first two elements of the reduced feature vectors from each placement position after processing with SRELM feature projection in Fig. 9. In contrast to the results from scatter plots before feature projection, the results after SRELM feature projection show that the cluster of each grasp type in each position is quite well separated. Moreover, the center of each cluster from each grasp type slightly changes when considering different object placement positions. The SI values from nine positions are in the range from 11.17 to 15.16. When we combine all clusters from all nine positions, the degree of separation still can be observed resulting in SI at 11.39 as shown in Fig. 5D.

The class separability of four projection techniques measured by the SI value in Table 3 shows that the supervised projections, namely LDA and SRELM, produce a higher SI compared to those from the unsupervised projections, namely PCA and t-SNE. While the largest SI 14.88 is from SRELM, PCA provides the smallest SI at 6.42. According to *Chu et al. (2007)*, the four different projection techniques LDA, PCA, nonlinear discriminant analysis (NLDA, a supervised nonlinear feature projection), and self-organizing feature map (SOFM, an unsupervised nonlinear feature projection) were applied to EMG pattern recognition when the limb positions were fixed. The class separability was measured by

Fisher's index. The results also indicated that the supervised methods (LDA and NLDA) outperformed the unsupervised methods (PCA and SOFM). While the highest Fisher's index 25,345.8 was from LDA, the lowest Fisher's index 0.0839 was from PCA.

## Classification accuracy

Figure 6 shows that variation in classification errors from the seven classifiers is quite low with the supervised feature projections (LDA and SRELM) compared to that from the unsupervised feature projections (PCA and t-SNE). For example, the classification errors from the seven classifiers using SRELM are in the range from 1.58% to 2.65% compared to those using t-SNE ranging from 1.27% to 17.15%. The classification error rate from using the t-SNE projection clearly varies with classifier type. It seems that t-SNE is not appropriate for linear classifiers, namely LDA and SVML, resulting in quite high classification errors (12.38–17.15%). However, better results are achieved when using the nonlinear classifiers. Therefore, when the unsupervised feature projection like t-SNE is applied, the classifier has to be carefully selected to achieve good classification error. The pairing between t-SNE and KNN is able to produce the minimum error rate of 1.27%. Nevertheless, there is no statistically significant difference compared with the pairing SRELM-KNN ($p = 0.462$). Then, it can refer that SRELM features give well separated cluster distribution in the projected features from different positions, better compared to the other feature projection techniques, resulting in no significant differences in classification error rate for the different classifiers.

Khushaba et al. (2014) proposed an alternative feature extraction based on time-dependent spectral analysis to resolve the variations in limb positions in the sagittal plane. The orthogonal fuzzy neighborhood discriminant analysis (OFNDA), which was a linear supervised feature projection, was employed for dimensionality reduction of EMG features. Eight types of hand motions from five different limb positions were classified when training with features from multiple limb positions. The classification errors for four classifiers including SVM, LDA, KNN ($k = 5$), and Extreme Learning Machine (ELM, 780 nodes in the hidden layer) were 8.85%, 9.62%, 9.34%, and 8.86%, respectively. We can see that the classification errors from OFNDA are also in a low variation, which is similar to our results. In other words, the supervised feature projection tends to provide less variation in classification error compared to the unsupervised feature projection. Similar results were reported by Khushaba, Al-Ani & Al-Jumaily (2010). When the projected features by OFNDA were classified with the linear (LIBLINEAR) and nonlinear (multilayer perceptron) classifiers, there was no significant difference in classification error ($p = 0.7069$).

In accordance with Chu et al. (2007), nine hand motions were classified by a multilayer perceptron, which was a nonlinear classifier, when the limb position was fixed. The classification error from four feature projections, namely LDA, PCA, NLDA, and SOFM, was 2.6%, 4.1%, 2.1%, and 3.8%, respectively. These results show that even classification error from PCA is in the same range with the other feature projections. However, our classification error from PCA is quite high compared to that from other feature projections. For example, when the NN classifier is used, our classification errors from

PCA and LDA feature projections are 13.58% and 3.35%, respectively. This may be caused by the differences in EMG data characteristic. In other words, while the EMG data in *Chu et al. (2007)* were acquired from fixed limb positions, the EMG data in this paper were recorded from variable limb positions.

## CONCLUSION

In this study, we evaluated feature projection techniques for an EMG pattern-recognition system to support myoelectric hand control, which is affected from variations in upper limb position. The four different projection techniques PCA (linear unsupervised), LDA (linear supervised), t-SNE (nonlinear unsupervised) and SRELM (nonlinear supervised) were evaluated with seven different classifier types to classify five different object grasps from nine positions in the transverse plane.

The results from feature visualization showed that SREML produced the remarkable best class separability, leading to a low classification error rate. From a classification accuracy comparison, the statistical analyses reveal that the choice of projection technique is a more significant factor than the choice of classifier type. The unsupervised projections, namely PCA and t-SNE, provide a wide range of classification performances depending on classifier type. In contrast, the supervised projections, namely LDA and SRELM, were able to achieve comparable performances regardless of classifier choice. Moreover, the nonlinear projections may offer an opportunity to achieve a high classification accuracy.

Regarding our results, the reduced feature vectors from SRELM showed the best performance not only in terms of achieving a high feature separability but also obtaining a low classification error rate (1.50–2.65%). It can be concluded that SRELM is able to provide an invariant feature and is efficient in representing the effects on limb position variation for myoelectric hand control.

From the present results, there are some further potential issues need to be addressed. Although the nonlinear projection techniques can promise good classification accuracy, they might spend more processing time than the linear projection techniques, especially when the original feature vector is of high dimensionality and variation. Consequently, the impacts of EMG features and the number of channels need to be studied in the future. Alternatively, a linear-nonlinear feature projection, a cascade of two projection which are linear and nonlinear, might be possible solution to enable the real-time system as had been reported by *Chu, Moon & Mun (2006)*. According to our present results, the minimum classification error is achieved by pairing t-SNE projection with KNN classifier, which are both unsupervised learning tools, resulting in non-mathematical mapping models (*i.e.*, relying on storing prior data, not only a small number of model parameters). This is quite difficult to implement in intuitively appealing control. For t-SNE, the corresponding mapping model might be indirectly constructed by employing some supervise learning such as artificial neural network as had been reported by *Oliveira, Machado & Andrade (2018)*. In the future, the knowledge from this study will be used with prosthetic devices to improve the quality of life for people who have had their upper limbs amputated.

## ACKNOWLEDGEMENTS

Also, the authors would like to thank the Research and Development Office (RDO), Prince of Songkla University and Associate Professor Dr. Seppo Karrila, Faculty of Science and Technology, Prince of Songkla University for commenting on this paper.

### Funding

This work was jointly funded by the Thailand Research Fund (TRF) through the Royal Golden Jubilee Ph.D. Program under Grant No. PHD/0095/2559 and in part by the Thailand Research Fund and Faculty of Engineering, Prince of Songkla University through Contract No. RSA6280016. The funders had no role in study design, data collection and analysis, decision to publish, or preparation of the manuscript.

### Grant Disclosures

The following grant information was disclosed by the authors:
Thailand Research Fund (TRF): PHD/0095/2559.
Thailand Research Fund and Faculty of Engineering, Prince of Songkla University: RSA6280016.

### Competing Interests

The authors declare that they have no competing interests.

### Author Contributions

- Nantarika Thiamchoo conceived and designed the experiments, performed the experiments, analyzed the data, performed the computation work, prepared figures and/or tables, authored or reviewed drafts of the paper, and approved the final draft.
- Pornchai Phukpattaranont conceived and designed the experiments, performed the experiments, analyzed the data, prepared figures and/or tables, authored or reviewed drafts of the paper, and approved the final draft.

### Ethics

The following information was supplied relating to ethical approvals (*i.e.*, approving body and any reference numbers):

The Human Research Ethical Committee of the Faculty of Medicine, Prince of Songkla University.

### Data Availability

The raw data are available in the Supplemental Files.

### Supplemental Information

Supplemental information for this article can be found online at http://dx.doi.org/10.7717/peerj-cs.949#supplemental-information.

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
