# Peer review of "Evaluation of feature projection techniques in object grasp classification using electromyogram signals from different limb positions"

_PeerJ Computer Science, doi:10.7717/peerj-cs.949_

## Round 0.1 · original submission · Minor Revisions

Thanks a trillion for submitting your paper to PeerJ Computer Science. We invite you to submit a revised paper after addressing all reviewers' comments. Also, make sure to thoroughly revise your manuscript to correct all typo/grammatical/spelling/formatting/style errors.

Reviewer 1 ·

Basic reporting

The paper is concerned with feature projection methods applied to the domain of EMG signal processing. In particular, data from several sensors is collected in order to classify grasps. Since the number of computed features may be large in such tasks, it is desired to reduce the dimensions before the application of classification algorithms. The authors test several methods for dimension reduction, some linear and some non-linear, with an emphasis on the non-linear technique named spectral regression extreme learning machine. This method was shown to have the best performance.

The level of English is good. The background and Introduction are adequate. Figures are of good quality.

Experimental design

The experimental designs is explained in detail, the presented figures are helpful. Some comments regarding the algorithmic implementation are detailed bellow.

Validity of the findings

Experimental results are adequate, and the findings are solid.

Additional comments

Overall, the topic is of interest, and the paper is solid, but I would say that the contribution mainly focuses on the question of comparison between dimension reduction techniques and a focus on the dataset. The fact that limb position changes in the dataset, is more challenging compared to some previous studies (still this seems like a not-very-hard classification task).

Comments:


Page 3, line 126 – This is more challenge  more challenging

Feature projection section:
• It is mentioned that the number of coordinates in the projected space is the number of grasp types minus one. It may be the case that more dimensions are needed for a better separation. Did you try other numbers?


Page 6, line 191 and on:
• The train-test separation was not clear to me. I assumed that the classification is not multi-subject, rather it is single subject (train-test belongs to one subject). Is that so?
• Does the train data contain all of the recorded positions? Or are some 'new' positions left out for the test data?
• It would be interesting to see the performance of the methods as the number of training samples is reduced.
• Please explain in more detail how the projected coordinates in each method are extended to the test data. In PCA, new points are normalized and projected to the train model. How about tSNE? It is not trivial to extend these coordinates. Is the model built over with the test data?
• Same question for SRELM.

Page 6 line 217: SRELM
• Since this method is new and probably most readers are not familiar with it, it would be useful to add a more detailed explanations of its construction, maybe in a pseudo code manner (instead of sending the reader to past papers).

Results section
• Please add an opening sentence before the sentence in line 250 (Page 8) that starts with "In Table 2…"

Reviewer 2 ·

Basic reporting

1. Subheadings must be bold, followed by a period, and start a new paragraph e.g. Background. The background section text goes here...
2. Figures are of poor resolution and clarity. They need to be revised with high resolution

Experimental design

1. I suggest that you improve the description about % split of train and test set at lines 191- 192. Also give system specifications.
2. Feature extraction methods should be improved with more elaboration and illustrations.
3. More clarity on the Classifier parameter selection should be included

Validity of the findings

1. No new hybrid algorithm/enhancement based on feature projection method is proposed.
2. No simulation/Pseudo-codes or algorithm is given.
3. Novelty of work should be strongly emphasized by authors.

Additional comments

1. Intro & background to show context are well explained. Problem statement is well defined
2. Literature well referenced & relevant
3. Rigorous investigation performed to a high technical & ethical standard.
4. Conclusions are well stated, linked to original research question & limited to supporting results

·

Basic reporting

No Comment

Experimental design

NO Comment

Validity of the findings

The authors have mentioned in their paper "In this study, the number of hidden nodes 1000 and alpha 1
223 were used. Both were determined by trial and error."
It would be better if the authors gave a range or reason for picking up a certain range and then coming to the experimental value.

Additional comments

1. The paper is written well and results show a varied range of Classification accuracy.
It would be interesting to comment on the reason for sometimes very low accuracy /high accuracy. Correlating the results with data distribution can help in this regard.
2. work with subjects needing prosthesis can also be a step for future work
3. The authors need to develop their own techniques / modify the existing for novelty in the paper.

---

## Round 0.2 · accepted · Accept

Congratulations on revising the paper and addressing all reviewers' comments. I am glad that your paper is now accepted for publication.

Reviewer 1 ·

Basic reporting

The paper was majorly improved.
All the point I raised were answered and addressed.
More experiments were preformed, especially these emphasize the influence of the train set size.
Additional text was added to explain various points that were not clear in the initial manuscript.

Experimental design

Adequate

Validity of the findings

Adequate

Reviewer 2 ·

Basic reporting

No comment

Experimental design

No comment

Validity of the findings

No comment

Additional comments

Authors have revised the manuscript as per guidelines. The proposed algorithms are added in the paper. The result section is very well written

·

Basic reporting

no comment

Experimental design

no comment

Validity of the findings

no comment

Additional comments

The paper includes all the suggestions given earlier. Clarity and Novelty aspects are covered in depth.